# Ultraviolet-C Irradiation, Heat, and Storage as Potential Methods of Inactivating SARS-CoV-2 and Bacterial Pathogens on Filtering Facepiece Respirators

**DOI:** 10.3390/pathogens11010083

**Published:** 2022-01-10

**Authors:** Rhodri Harfoot, Deborah B. Y. Yung, William A. Anderson, Cervantée E. K. Wild, Nicolene Coetzee, Leonor C. Hernández, Blair Lawley, Daniel Pletzer, José G. B. Derraik, Yvonne C. Anderson, Miguel E. Quiñones-Mateu

**Affiliations:** 1Department of Microbiology & Immunology, School of Biomedical Sciences, University of Otago, Dunedin 9016, New Zealand; rhodri.harfoot@otago.ac.nz (R.H.); yunde607@student.otago.ac.nz (D.B.Y.Y.); leonor.hernandez@otago.ac.nz (L.C.H.); blair.lawley@otago.ac.nz (B.L.); daniel.pletzer@otago.ac.nz (D.P.); 2Department of Chemical Engineering, University of Waterloo, Waterloo, ON N2L 3G1, Canada; wanderson@uwaterloo.ca; 3Department of Paediatrics, Child and Youth Health, University of Auckland, Auckland 1010, New Zealand; cervantee.wild@auckland.ac.nz (C.E.K.W.); nicolene.coetzee@auckland.ac.nz (N.C.); j.derraik@auckland.ac.nz (J.G.B.D.); 4Webster Centre for Infectious Diseases, University of Otago, Dunedin 9016, New Zealand

**Keywords:** SARS-CoV-2, COVID-19, personal protective equipment, PPE, disinfection, bacteria, UV-C, New Zealand

## Abstract

The arrival of SARS-CoV-2 to Aotearoa/New Zealand in February 2020 triggered a massive response at multiple levels. Procurement and sustainability of medical supplies to hospitals and clinics during the then upcoming COVID-19 pandemic was one of the top priorities. Continuing access to new personal protective equipment (PPE) was not guaranteed; thus, disinfecting and reusing PPE was considered as a potential alternative. Here, we describe part of a local program intended to test and implement a system to disinfect PPE for potential reuse in New Zealand. We used filtering facepiece respirator (FFR) coupons inoculated with SARS-CoV-2 or clinically relevant multidrug-resistant pathogens (*Acinetobacter baumannii* Ab5075, methicillin-resistant *Staphylococcus aureus* USA300 LAC and cystic-fibrosis isolate *Pseudomonas aeruginosa* LESB58), to evaluate the potential use of ultraviolet-C germicidal irradiation (UV-C) or dry heat treatment to disinfect PPE. An applied UV-C dose of 1000 mJ/cm^2^ was sufficient to completely inactivate high doses of SARS-CoV-2; however, irregularities in the FFR coupons hindered the efficacy of UV-C to fully inactivate the virus, even at higher UV-C doses (2000 mJ/cm^2^). Conversely, incubating contaminated FFR coupons at 65 °C for 30 min or 70 °C for 15 min, was sufficient to block SARS-CoV-2 replication, even in the presence of mucin or a soil load (mimicking salivary or respiratory secretions, respectively). Dry heat (90 min at 75 °C to 80 °C) effectively killed 10^6^ planktonic bacteria; however, even extending the incubation time up to two hours at 80 °C did not completely kill bacteria when grown in colony biofilms. Importantly, we also showed that FFR material can harbor replication-competent SARS-CoV-2 for up to 35 days at room temperature in the presence of a soil load. We are currently using these findings to optimize and establish a robust process for decontaminating, reusing, and reducing wastage of PPE in New Zealand.

## 1. Introduction

SARS-CoV-2 was first identified in Aotearoa/New Zealand (henceforth referred to as New Zealand) in February 2020 [1]. A combination of public-health measures including a swift lockdown [1], establishment of diagnostic capabilities [2,3,4], and research programs [5,6], together with an exemplary response from the community, allowed the country to control the initial virus outbreak [1,5,7]. Even to date, in the middle of surges of SARS-CoV-2 delta and omicron variants worldwide, New Zealand is one of the countries with the lowest number of confirmed COVID-19 cases per capita (https://nzcoviddashboard.esr.cri.nz/#!/international, accessed on 26 November 2021). However, during the initial months of the COVID-19 pandemic, global supply shortages with increased demand and interrupted supply chains resulted in shortages of critical hospital equipment, including personal protective equipment (PPE) [7]. In most healthcare settings, PPE includes masks (i.e., surgical masks and filtering facepiece respirators (FFRs)), eye protection (face shields and glasses), and gowns and gloves (https://www.who.int/publications/m/item/how-to-guide-putting-on-ppe, accessed on 26 November 2021). As PPE shortages increased, healthcare personnel had to consider reusing otherwise-disposable materials, including PPE. PPE reuse has been widely reported by healthcare workers [8]. The global stock of PPE, including supply chains, was compromised to the point that it was difficult for many healthcare workers to have access to PPE, requiring the World Health Organization to produce guidance as to the rational use of PPE in severe shortages (https://www.bbc.co.uk/news/health-52145140, accessed on 11 October 2021). Calls were made for ideas to conserve the supply of PPE [9]; thus, saving, disinfecting and reusing PPE was considered as one potential solution within the healthcare-worker research community [10].

Even prior to the COVID-19 pandemic, there was concern about the sustainability of PPE supply in the event of a major transmissible disease outbreak [11]. Multiple studies have evaluated the efficacy of a variety of chemical, thermal, and germicidal ultraviolet-C irradiation (UV-C) disinfection methodologies to disinfect FFRs or similar materials [12,13,14,15,16,17,18,19]. The relative merits of these and other methods (including ‘storage’) have been recently reviewed [20]. Ultraviolet germicidal irradiation (UVGI, 250 to 280 nm) is a portion of the UV-C spectrum (200 to 280 nm) frequently generated using low-pressure mercury lamps emitting at 253.7 nm [21]. Although the dose (or fluence) required to inactivate bacteria and viruses has been studied over the years [21], the current COVID-19 pandemic has reignited the need to develop cost-effective procedures to decontaminate surfaces and materials using UV-C [22,23,24,25,26,27,28,29,30,31,32,33,34,35,36,37,38,39,40]. In the case of SARS-CoV-2, UV-C doses ranging from 16.9 mJ/cm^2^ in cell culture [41] to 1500 mJ/cm^2^ on FFR [42] are reportedly required to fully inactivate virus replication. However, any multi-dimensional surface (such as a mask) may not be equally irradiated across all its surfaces. Similarly, numerous studies have assessed the effect of heat treatment to inactivate SARS-CoV-2 [20]. While a range of temperatures and times has been described to inactivate many viruses [20], coronaviruses—like other enveloped viruses—are usually completely inactivated at 56 °C for 45 min, 60 °C for 30 min, 65 °C for 15 min, or 80 °C for 5 min [20]. However, heat treatment is affected by multiple variables, such as humidity, viscosity, and even protein content [20]. Therefore, due to the wide variability of conditions and PPE, including types of FFR models, disinfection methods need to be evaluated and validated to replicate real-world conditions in different geographic areas.

In addition to SARS-CoV-2 and other viral contaminations on PPE, Delanghe et al. [43] recently investigated bacterial contamination on surgical face masks in the community. They showed massive contamination of more than 10,000 colony-forming units (CFU) per surgical mask with >40% drug resistant bacteria including *Staphylococcus*, *Acinetobacter*, and *Bacillus*. The situation in the clinical environment is similar to that in the community, with a recent study showing that 36% of PPE-wearing healthcare workers were contaminated with multidrug-resistant pathogens upon patient contact [44]. The main clinically important pathogens in hospital settings are methicillin-resistant *S. aureus* (MRSA), vancomycin-resistant *Enterococcus*, and multidrug-resistant *Pseudomonas aeruginosa* as well as *A. baumannii*.

This study was the first of a comprehensive multidisciplinary program aimed at implementing and testing a mobile, scalable solution for disinfection of PPE for potential reuse in New Zealand and beyond. The final goal of our overall program was to create a solution to increase the supply of PPE for users at the frontline (particularly healthcare workers), mitigating potential disruptions to PPE supply, and reducing environmental waste from single-use PPE. Here we assessed the use of UV-C irradiation or dry heat treatment to inactivate not only SARS-CoV-2 but also opportunistic pathogens responsible for nosocomial infections that could contaminate FFRs. We also evaluated the stability of SARS-CoV-2 over time in the presence or absence of conditions mimicking salivary or respiratory secretions. 

## 2. Results

### 2.1. Testing FFR Coupons with IAV and SARS-CoV-2 

Prior to the evaluation of UV-C irradiation or dry heat treatment to inactivate SARS-CoV-2 replication, we tested the ability to “contaminate” FFR coupons with viral particles and rescue replication-competent virions from this material. We first used a virus that could be manipulated in a Physical Containment 2 (PC2) laboratory in case we needed to address any methodological issues. FFR coupons were inoculated with serial dilutions of the IAV A/Mallard/Alberta/287/2012 (H1N1) strain in duplicate, incubated for one hour, with inverted FFR coupons then placed into wells containing ELVIRA^®^ Flu A cells (Appendix A). This reporter cell line expresses the firefly luciferase gene in response to infection with IAV [45], facilitating the quantification of viral replication. As shown in Appendix A, the amount of luciferase produced was proportional to the amount of IAV deposited in each FFR coupon, although a fraction of the same amount of virus was used to directly infect the ELVIRA^®^ Flu A cells as positive control. The same approach was used with SARS-CoV-2 and Vero cells in the PC3 facility. The level of cell death (CPE) due to viral infection correlated with the amount of virus used to inoculate the FFR coupons (Appendix A). Importantly, the FFR coupon by itself had no effect on cell viability as shown in the wells with either ELVIRA^®^ Flu A or Vero cells alone. 

### 2.2. UV-C Irradation Inactivates SARS-CoV-2 but Its Efficacy Is Hampered by Surface Irregularities 

Following the proof-of-concept experiment using IAV and SARS-CoV-2 with the FFR coupons, we assessed the ability of UV-C radiation to disinfect FFRs inoculated with SARS-CoV-2. We used a UV-C irradiation chamber designed and constructed for this study (Appendix A) to expose the inoculated coupons to different doses of UV-C irradiation. As described in Materials and Methods, the FFR coupons were stitched together to prevent the separation of the multiple layers, and irradiated with doses ranging from 0 to 2000 mJ/cm^2^. Although the negative (FFR with no virus) and positive (untreated FFR plus SARS-CoV-2) controls showed no and full CPE, respectively, no consistent effect was observed with the UV-C-irradiated FFR coupons, even at the highest dose of 2000 mJ/cm^2^. This was evident in the FFR coupons exposed to SARS-CoV-2 in a cell-culture-medium solution, where some of the four replicates showed clear signs of viral replication. This lack of UV-C inactivation was somewhat exacerbated by the presence of mucin or soil load in the virus solution (Figure 1).

To investigate the possibility that uneven distribution of UV-C radiation across the chamber could be responsible for the inconsistency inactivating SARS-CoV-2, we used chemical actinometry to quantify the actual UV-C dose received at each position of the 24-well plate lid within the UV-C irradiation apparatus. Limited variability in UV-C dose was observed among the positions, with a median of 692 mJ/cm^2^ (range 669 to 722 mJ/cm^2^) and coefficient of variation of 1.9% (Figure 1B). 

Since the position in the UV-C chamber and radiation dose did not explain the variable SARS-CoV-2 inactivation, we explored the possibility that the stitches in FFR coupons were contributing to inconsistent UV-C disinfection results. Unstitched FFR coupons were therefore generated, minimizing manipulation to avoid the separation of the different layers. Similar to the stitched FFR coupons, these were inoculated with SARS-CoV-2 in the outer surface and exposed to UV-C radiation. As shown in Figure 2A, UV-C doses of 1000 mJ/cm^2^ or greater resulted in complete SARS-CoV-2 inactivation in all four replicates. These results were corroborated in an experiment where both stitched and unstitched FFR coupons were inoculated with SARS-CoV-2 and simultaneously exposed to an UV-C dose of 1500 mJ/cm^2^. The virus in all unstitched FFR coupons was completely inactivated, with high variability in the stitched FFR coupons, showing full inactivation in some and detecting SARS-CoV-2 replication in other FFR coupons, regardless of their position in the UV-C chamber (Figure 2B). Thus, UV-C irradiation inactivated SARS-CoV-2 replication in flat, unaltered FFR coupons but failed to reproducibly inactivate the virus in irregular FFR material.

### 2.3. Dry Heat Inactivates SARS-CoV-2 and Planktonic Bacteria but Not Bacterial-Colony Films 

Given the variable UV-C findings as a disinfection method, we explored the efficacy of dry heat to disinfect FFRs inoculated with SARS-CoV-2 or bacteria. Irregular (stitched) FFR coupons were inoculated with SARS-CoV-2 in cell-culture medium, with mucin or soil, then incubated for 15 to 90 min at room temperature (mean 22 °C, range 19 °C to 24 °C), 60 °C, 65 °C, or 70 °C. All the stitched FFR coupons incubated at room temperature, regardless of the condition (medium, mucin, or soil) or time, harbored replication-competent SARS-CoV-2 (Figure 3). However, incubation at 65 °C for 30 min or 70 °C for 15 min, was sufficient to fully inactivate SARS-CoV-2 replication, even in the presence or mucin or soil (Figure 3).

Since dry heat successfully inactivated SARS-CoV-2 from stitched FFR coupons, we adapted the method to test the killing efficacy of dry heat against clinically-relevant bacteria that could contaminate filtering facepiece respirators, i.e., pathogenic, multidrug resistant *A. baumannii* Ab5075, methicillin-resistant *S. aureus* USA300 LAC, and cystic fibrosis isolate *P. aeruginosa* LESB58. Dry heat effectively killed 1 × 10^6^ CFU of *S. aureus* and *A. baumannii* in PBS or mucin in 90 min at 80 °C, while 90 min at 75 °C was sufficient to kill similar bacterial numbers of *P. aeruginosa* (Figure 4). To further investigate heavy contamination of PPE, we used bacterial colony biofilms. The number of bacteria within a biofilm represented by one colony averaged from ~1.56 × 10^8^ CFU (*P. aeruginosa*) to ~3.26 × 10^7^ CFU (*S. aureus*) and ~4.4 × 10^8^ CFU (*A. baumannii*). Even two hours at 80 °C failed to completely kill any of the bacteria within the biofilm (Figure 4). In addition, a stand-down experiment where bacterial biofilms were incubated in the FFR coupons for 24 h prior to treatment with dry heat, not only did not affect bacterial numbers in biofilms but bacterial biofilms survived all dry heat treatments (data not shown).

### 2.4. SARS-CoV-2 Remains Replication-Competent in FFR 

Multiple studies have evaluated the probability of SARS-CoV-2 persisting in different surfaces while maintaining its ability to infect and replicate in human cells [20,46]. Here we inoculated FFR coupons with SARS-CoV-2 and stored them at room temperature for up to 35 days to ascertain whether ‘storage’ may be a viable alternative to heat or UV-C disinfection methods. The relative humidity in the PC3 laboratory, where the contaminated FFR coupons were stored, was monitored and ranged from 19 to 34% (median 25%, Figure 5A). Replication-competent virus on cell-culture medium was detected up to 9 days post-inoculation; however, virus in mucin was still detected at day 14. Importantly, SARS-CoV-2 in the soil-load solution remained replication-competent in FFR for up to 35 days (Figure 5B). 

## 3. Discussion

SARS-CoV-2 arrived in late 2019, and within a few months resulted in a global health emergency [47]. PPE was in short supply across the globe, affecting the availability of this critical safety equipment in many countries, including New Zealand. The country was able to organize an good response to the COVID-19 pandemic, starting with a series of public health-measures [48,49], establishing molecular diagnostic assays [2,3,4], monitoring the SARS-CoV-2 variants in the region [6,50], and isolating the virus circulating in New Zealand [51]. However, New Zealand experienced the same PPE shortages as numerous other countries, and the NZ Office of the Auditor-General reported gaps in planning of PPE procurement and distribution, with insufficient national stock reserves (https://oag.parliament.nz/, accessed on 26 October 2021). In this study, we first established that UV-C fully inactivated SARS-CoV-2 on flat surfaces, although the irregularities of PPE, as replicated by the stitches in FFR coupons, seem to discard the feasibility of using UV-C as a reliable method of FFR disinfection. Second, we showed that dry heat successfully inactivated SARS-CoV-2 on FFR coupons at 65 °C for 30 min, even in the presence of mucin and soil load that mimicked saliva and respiratory secretions, respectively. However, if other potential bacterial pathogens present on PPE are considered, 90 min at 80 °C was required in the absence of bacterial biofilm. Finally, the practice of storing FFRs potentially contaminated with SARS-CoV-2 for ongoing reuse that seems to be common worldwide among the general public [52] and health professionals [53] is not an entirely safe practice, with replication-competent virus still present in coupons 14 or 35 days post-inoculation in the presence of mucin or soil load, respectively. 

Ultraviolet germicidal irradiation (UVGI) has been used to disinfect surfaces and PPE [21], for some time, including testing coupons cut from FFRs inoculated with bacteria or viruses [20,24,54,55,56]. The COVID-19 pandemic revitalized the need to use different methodologies to disinfect surfaces and material potentially contaminated with SARS-CoV-2 [20], with reported claims that UV-C irradiation is a rapid and cost-effective solution for PPE disinfection [57,58]. Numerous studies have described the use of UV-C irradiation to inactivate SARS-CoV-2 [44,59,60]. For example, a UV-C irradiance of 3.7 mJ/cm^2^ was sufficient to reduce more than three logs SARS-CoV-2 replication [41], while 16.9 mJ/cm^2^ resulted in complete viral inactivation in aqueous suspension [41]. Here, we verified that irradiating FFR material—contaminated with a relatively high SARS-CoV-2 load (5 × 10^4^ TCID_50_/mL)—with an estimated applied dose of 1000 mJ/cm^2^ was enough to completely inactivate viral replication. Unfortunately, any irregularity or alteration in the integrity of the material, as evidenced by the stitches used to keep the multiple layers of the facemask together, affected the ability of UV-C irradiation to consistently inactivate the virus, even at higher applied doses (2000 mJ/cm^2^). It is important to note that although the viral droplet was always placed on the external face of the FFR coupons, away from the stiches, we cannot rule out that in certain cases the virus solution could have been in contact with the stitching, favoring the transference of the virus to the inner layers, shielding it from the UV-C irradiation or even shadowing the virus inoculum by the stitching itself. Most studies have tested unaltered and smooth FFR coupons, undisturbed by seams, strap attachments, or other common geometric features proper of any regular face mask [17,20,30]. Some studies have assessed the impact of multiple rounds of UVGI on pressure drop and particle-filtration efficiency [20], as well as multiple reuses or extended FFR wear on fit and filtration [59]; however, they have not evaluated disinfection efficacy. This work highlights the fact that due to the multiple cracks and crevices of FFRs, UV-C disinfection may not be a reliable method for pathogen inactivation on such materials. 

Dry heat has also been evaluated as a potential method to inactivate viruses on contaminated FFRs for potential reuse [20]. Here we demonstrated that heating FFR material at 70 °C for as short as 15 min (or 65 °C for 30 min) was enough to fully inactivate high doses of SARS-CoV-2, even in irregular stitched FFR coupons where UV-C irradiation failed to eliminate replication-competent viruses. We also attempted to reproduce real-life conditions by re-suspending SARS-CoV-2 in two different solutions: an inoculum containing mucin, which is the main protein found in human saliva [60], and a soil load based on ASTM E2197-11 standards [61], to mimic conditions found in respiratory secretions. Interestingly, the presence of mucin, but not soil, somewhat masked the effect of dry heat treatment at low temperatures, allowing some SARS-CoV-2 to replicate.

Since PPE will most likely also be contaminated with other pathogens, e.g., bacteria, we tested the ability to kill bacterial species representative of non-spore-forming potential human pathogens. Recently it has been shown that *S. aureus* viability on PPE was reduced (>4 log_10_) by dry heat treatment for 90 min at 70 °C [62]. The authors used a very high bacterial inoculum (1–5 × 10^9^ CFU/mL) that was placed on cellulose-membrane filters mounted on top of N95 masks, not onto the N95 masks directly. In comparison, we used 90 min at 75 °C to effectively kill >10^6^
*S. aureus* from the FRR coupons directly. In addition, 90 min at 75–80 °C dry heat could also be used to effectively kill Gram-negative pathogens *A. baumannii* and *P. aeruginosa* when inoculated at a high density on FRR coupons. Biofilms found on surfaces in hospitals and healthcare facilities are an important source of bacterial contamination and transmission [63]. Most microorganisms produce biofilms for environmental protection from e.g., desiccation and must therefore be considered when developing new methods for the re-use of PPE. Our results highlight that heat treatment up to 120 min at 80 °C was insufficient to kill >10^7^ CFU of *S. aureus*, *A. baumannii,* or *P. aeruginosa* colony biofilms. Similarly, a recent study showed that *S. aureus* dry surface biofilms (1 × 10^7^ CFU) were resistant to dry heat treatment even at 100 °C for 60 min [64]. At this temperature the integrity of the FRR coupons would be compromised. These findings are important since in real-life, most bacteria grow in biofilms [65,66] and we expect that this will be the case in heavily soiled PPE, which is not suitable for reuse for this reason. Mask and PPE reprocessing should only be for visually clean and undamaged articles—otherwise it is imperative to discard them. Other than discarding soiled PPE, further studies are necessary to explore the best treatment to completely eliminate viable bacterial biofilms from soiled material using heat treatment. 

Finally, much has been written about the stability of viral particles in surfaces [20,46]. This was the focus of many studies and extensive debate early in the COVID-19 pandemic, i.e., how long would SARS-CoV-2 remain replication-competent in any given environment? There are a number of parameters to consider when determining viability, such as viral load, relative humidity, and temperature [20]. However, a limited number of studies have demonstrated the retrieving of replication-competent SARS-CoV-2 virions from PPE after 21 days [46]. Here we tested the stability of SARS-CoV-2 particles in the presence and absence of mucin and soil load over time. While the mucin medium had limited protective effect, the SARS-CoV-2 particles were able to remain replication-competent for more than a month in soil load, a solution mimicking respiratory secretions. It is possible that the additional protein content protects virions from dehydration. Since SARS-CoV-2 droplets will most likely be comprised of a mixture of virions, bacteria, and secretions from the upper respiratory tract, these findings showcase the need to thoroughly decontaminate PPE prior to being reused.

This study had a few limitations. We first acknowledge the apparent feasibility of other potential disinfection methods (e.g., vaporized hydrogen peroxide for PPE [67]); however, such methods could not be readily adopted in many areas, due to the need for supply of the chemical and the necessary health and safety issues required to handle it. There is some evidence that non-steam moist heat treatment may be more effective for virus inactivation in FFRs than dry heat. However, dry heat treatment involves a much simpler protocol that would be markedly easier to replicate independently of scale or context, particularly at locations where funds are limited and/or access to more specialized equipment is difficult. Second, the direct inoculation of large amounts of SARS-CoV-2 or bacteria on FFR coupons may not have accurately reproduced the droplet and/or aerosol contamination of PPE in real-world situations, particularly the potential penetration of droplets into FFR crevices, folds, and inner layers. We may have also used inoculums with higher than normal SARS-CoV-2 and bacteria loads, most likely overestimating the volume of these pathogens that could contaminate FFRs in practice, perhaps overwhelming the capacity of UV-C irradiation and/or dry heat to inactivate them. On the other hand, we were able to demonstrate that even under these conditions, dry heat treatment is able to inactivate SARS-CoV-2 and kill bacteria from contaminated FFRs. Finally, we acknowledge the importance of spore-forming pathogens in the proposed scenario of PPE disinfection for potential reuse, and this is currently being studied (yeasts were not able to be studied in New Zealand but also are important to consider). From a practical perspective, clear policies regarding immediate discarding of PPE used with patients with vomiting or symptoms suggestive of pathogens such as *Clostridium difficile*, and not using reused PPE when caring for immunocompromised patients would be imperative.

In conclusion, dry heat is able to disinfect PPE from SARS-CoV-2 and free-living bacterial pathogens (although not bacteria growing in colony biofilms) for potential reuse, addressing the safety of healthcare workers in the event of potential supply shortages, and the vast environmental impact of increased PPE waste across the globe. Further research is underway to understand the material properties of FFRs and other items of PPE after heat treatment, and the impact of wear between cycles.

## 4. Materials and Methods

### 4.1. Cells and Viruses

MDCK (CCL-34™ ATCC) and Vero (CCL-81™ ATCC) cells, a gift from Dr. Matloob Husain, University of Otago, were grown in DMEM (Gibco Thermo Fisher Scientific, Waltham, MA, USA) supplemented with 10% fetal bovine serum (FBS, Cellgro Mediatech, Manassas, VA, USA), 100 units/mL of penicillin, and 100 μg/mL of streptomycin (Gibco Thermo Fisher Scientific). VeroE6/TMPRSS2 [68] cells were purchased from the Japanese Collection of Research Bioresources Cell Bank (Osaka, Japan) and maintained as described above for Vero cells with the addition of 1 μg/mL of Geneticin™ (Gibco Thermo Fisher Scientific). ELVIRA^®^ Flu A cells [45] (Quidel Corporation, San Diego, CA, USA) were grown as described above for Vero cells with the addition of 150 μg/mL of hygromycin B (Gibco Thermo Fisher Scientific). All three cell lines were maintained in a humidified incubator with 5% CO_2_ at 37 °C. The influenza A virus (IAV) A/Mallard/Alberta/287/2012 (H1N1) strain was donated by Dr. Richard J. Webby (St. Jude Children’s Research Hospital, Memphis, TN, USA). The New Zealand SARS-CoV-2 NZ/Queenstown/01 strain was originally isolated by our group [51]. Virus stocks were prepared by growing IAV and SARS-CoV-2 in MDCK and Vero cells, respectively. Cell-culture supernatant was harvested, clarified by centrifugation at 1500 rpm, filtered through a 0.45 mm steriflip filter (Merck Millipore, Burlington, MA, USA), aliquoted, and stored at −80 °C until further use. Tissue culture dose for 50% infectivity (TCID_50_) was determined in triplicate for each serially diluted virus using the Reed and Muench method [69] and viral titers expressed as infectious units per milliliter (IU/mL).

### 4.2. Bacterial Strains and Growth Conditions 

The three bacterial multidrug-resistant strains used in this study were *Staphylococcus aureus* USA300 LAC [70], *Acinetobacter baumannii* Ab5075 [71], and *Pseudomonas aeruginosa* LESB58 [72]. All bacterial cultures were grown at 37 °C in tryptic soy broth (TSB; BD Difco™, BD Diagnostic Systems, Heidelberg, Germany) for 16 to 18 h. Bacterial colony biofilms were grown on Congo red agar (CRA) composed of BD Difco™ brain heart infusion broth (BHI; 37 g/L), sucrose (5 g/L), 1% agar, and Congo red dye (0.8 g/L) [73]. Biofilms were grown for 48 h (*A. baumannii* and *S. aureus*) or 96 h (*P. aeruginosa*). One colony of each bacterial species (*A. baumannii* colony size ~3 mm^2^, *S. aureus* ~1.5 mm^2^, and *P. aeruginosa* ~1.5 mm^2^) was selected per experiment. After the dry heat experiment, bacteria were cultured on either BHI + 5% sucrose + 20 μg/mL chloramphenicol (Sigma-Aldrich, St. Louis, MO, USA) to select for *A. baumannii*, 7.5% NaCl or 15 μg/mL colistin (Sigma) to select for *S. aureus*, or Pseudomonas cetrimide agar (PCA; Oxoid™) to select for *P. aeruginosa*.

### 4.3. Mucin and Soil Media

A 3 mg/mL solution of mucin from bovine submaxillary glands (Merck Sigma-Aldrich, Darmstadt, Germany) was prepared in 1× phosphate-buffered saline (PBS) [74]. Soil load was also prepared following ASTM standards [61]; i.e., 0.5 g of tryptone (Merck Sigma-Aldrich), 0.5 g of bovine serum albumin (BSA, Merck Sigma-Aldrich), and 0.04 g of bovine mucin (Merck Sigma-Aldrich) in 10 mL of PBS, sterilized by passage through a 0.22 µm filter and stored at −20 °C.

### 4.4. FFR Coupon Preparation

Circular coupons were created from the same FFR model (Help-It FFP2 respirators, QSi, Whanganui, New Zealand). As FFRs are multi-layered, individual coupons were stitched using sewing machines (Singer 4432 Heavy Duty, La Vergne, TN, USA; and Elna 3003, Geneva, Switzerland) to secure layers in place during experiments. Eight-millimeter coupons were then cut using an alcohol-disinfected drive pin punch. Only synthetic threads (100% polyester; Gütermann, Gutach-Breisgau, Germany) were used, as cotton threads are more hydrophilic and could potentially absorb liquid from the inoculum. Briefly, both surfaces of the coupon were stitched at the same time, with a dark-colored thread marking the outer-facing (external) surface of the FFR and a light-colored thread on the wearer-facing (internal) side (Appendix A).

### 4.5. Coupon Inoculation 

Single FFR coupons were placed either on the lid (UV-C treatment) or into individual wells (dry heat treatment and storage) of 24-well cell-culture plates (Greiner Bio-One, Kremsmünster, Austria), with the external surface marked with the dark-colored thread facing upwards, and inoculated with 5 µL of either virus or bacteria in a solution of medium, mucin, or soil load. After each respective treatment, as described below, coupons were lifted out of the plates with sterile forceps and used to inoculate the respective cell-culture or bacterial-growth medium.

### 4.6. Ultraviolet-C Irradiation Treatment 

A UV-C irradiation chamber was manufactured specifically for this study (Sean McNulty, UV Solutionz, Kerikeri, New Zealand), consisting of a metal box 50 cm long × 23.5 cm wide × 26 cm high, sized to be placed inside a Class II Biosafety Cabinet in the Physical Containment 3 (PC3) facility at the University of Otago (Dunedin, New Zealand) (Appendix A). It contained a single drawer (27 cm long × 10 cm wide × 5 cm deep) designed to accommodate a 24-well cell culture plate (Appendix A). The box contained a single UV lamp model EE4066LP with an amalgam filament, 1.8 Amp, 40 V, and 72 W (UV Solutionz, Kerikeri, New Zealand). The average UVC irradiance was 5.56 mW/cm^2^ at 254 nm, within a narrow range of ±0.10 mW/cm^2^, as assessed over a six-minute period by a calibrated radiometer (Tiny Tracker; Opsytee Dr. Gröbel, Ettlingen, Germany). The lamp was located 100 mm from the base of the drawer, where the sensor eye was placed (Appendix A). FFR coupons were exposed to progressively greater applied doses of UVC, ranging from 250 to 2000 mJ/cm^2^, calculated as the product of the lamp’s irradiance (E_e_, in mW/cm^2^) over time (in seconds), i.e., D = E_e_ × time [21].

Although the FFR coupons were initially to be placed within the wells of a sterile 24-well cell-culture plate (CELLSTAR, Greiner Bio-One GmbH, Frickenhausen, Germany), preliminary tests with the UV-C chamber indicated that due to the depth of the wells, there was a shadowing effect that could interfere with the UV-C irradiance of each individual coupon. Therefore, the FFR coupons were instead placed on the flat lids of each well plate, where slightly raised circles marked the positions of the corresponding wells. In addition, no FFR coupons were placed at the positions located in the two outer rows of the lids to limit the variation in the distance between the FRR coupons and the lamp, reducing the number of FFR coupons to be tested at any given time to 16 per plate lid. Each FFR coupon was inoculated with 5 µL of the SARS-CoV-2 NZ/Queenstown/01 strain (10^7^ TCID_50_/mL) and incubated for 10 min. Constant UV-C irradiance was achieved 5 min after turning on the UV-C lamp, and at that time, the plate lid with the corresponding FFR coupon was placed in the center of the drawer (Appendix A). The time of UV-C exposure was controlled using a shutter. A data logger (EL-USB-2-LCD, Lascar Electronics, Hong Kong, China) was also placed within the UV-C chamber to monitor temperature and relative humidity. The UV-C chamber was placed inside a biosafety cabinet, located in the PC3 facility for the duration of the study. 

### 4.7. Chemical Actinometry 

To validate the radiometer measurements of the UV-C irradiation chamber, as well as verify that the UV-C radiation was equally distributed across all surfaces of the FFR coupons, we used an iodide–iodate chemical actinometer method adapted from Rahn et al. [75,76]. In this method, a photochemical reaction product (triiodide) is produced in proportion to the number of UV-C photons absorbed in the solution, and the concentration of triiodide is measured spectrophotometrically in the visible spectrum. The moles of triiodide formed are then used to determine the photon flux at the surface of the solution as well as the UV-C fluence (dose). Here, a mixture of KI (0.6 M) and KIO_3_ (0.1 M) was prepared in a 0.01 M borax buffer at pH 9.25. Each well in the 24-well plate was filled to the brim with the actinometer solution, and the plate was exposed to UV-C irradiation in the device for a specific period of time using the shutter mechanism after lamp warm-up. The actinometer solution strongly absorbs at 254 nm, resulting in approximately 99.99% photon absorbance within the top 0.5 mm of solution in the well. The absorbance at 480 nm was then measured in each well using a plate reader (VICTOR Nivo, PerkinElmer, Waltham, MA, USA). The molar absorption coefficient for triiodide at 480 nm was determined to be 526 M^−^^1^cm^−^^1^ based on a second-order polynomial small extrapolation from the values reported at 426, 450, and 476 nm [76]. Using the reported quantum yield of 0.73 moles of triiodide formed per mole of photons absorbed, Planck’s constant, the volume of actinometer solution, and the cross-sectional area of each well, the UVC dose or fluence (mJ/cm^2^) was determined for each well at each total exposure time.

### 4.8. Dry Heat Treatment 

Individual FFR coupons were placed within the wells of 24-well plates and inoculated with either (i) 5 µL of the SARS-CoV-2 NZ/Queenstown/01 strain (10^7^ TCID_50_/mL) or (ii) 5 μL of *S. aureus, A. baumannii*, or *P. aeruginosa* from bacteria culture (1 × 10^6^ CFU) or biofilm (one colony representing 3 × 10^7^ to 5 × 10^8^ CFU). Following a 10 min incubation, the 24-well plate with the FFR coupons was placed on a pre-heated Eppendorf ThermoMixer^®^ C with a SmartBlock™ (Eppendorf, Hamburg, Germany) and covered with the ThermoTop^®^ (Eppendorf) to guarantee a constant thermal profile over the entire plate, at different temperatures (22 to 80 °C) and times (15 to 120 min). Finally, we also performed a 24 h stand-down experiment where bacterial biofilms were prepared as above but left for 24 h at room temperature prior to starting the experiment. Control experiments to determine the number of bacteria within the biofilm were determined by the selection of one colony resuspended in 1 mL PBS, diluted and plated onto the respective selective media for each strain (five colonies each), incubated overnight, and colonies counted.

### 4.9. SARS-CoV-2 Stability in FFR Coupons 

Individual FFR coupons were placed within the wells of 24-well plates and inoculated with 5 µL of the SARS-CoV-2 NZ/Queenstown/01 strain (10^7^ TCID_50_/mL). Plates were then stored at room temperature for 2 to 35 days, inside a biosafety cabinet in the PC3 facility after having been wrapped in aluminum foil to avoid virus inactivation due to the UV irradiation from the biosafety cabinet. 

### 4.10. SARS-CoV-2 Replication and Bacterial Growth Determination 

After the different UV-C or dry heat treatments, as well as the stability study, the FFR coupons were removed from each plate lid or 24-well plate with sterile forceps and the ability of the virus and bacteria to replicate evaluated as follows:

#### 4.10.1. SARS-CoV-2

FFR coupons were inverted so that the inoculated face (dark-colored thread facing downwards) was in contact with the cell culture media of a matching 24-well plate containing 50,000 VeroE6/TMPRSS2 cells/well. Cells were incubated at 37 °C, 5% CO_2_ for 72 h and SARS-CoV-2 replication quantified by determining cytopathic effect (CPE), as described [51]. 

#### 4.10.2. Bacteria

FFR coupons were transferred to 1.5 mL microcentrifuge tubes with 500 μL of PBS, vortexed for 30 s, and the liquid with the coupon spread onto selective agar plates. Plates were incubated for 48 h on selective plates at 37 °C before visual growth was recorded. Each experiment was performed with two biological replicates consisting of six technical replicates (FFR coupons) for 90 min and 120 min, respectively.

## Figures and Tables

**Figure 1 pathogens-11-00083-f001:**
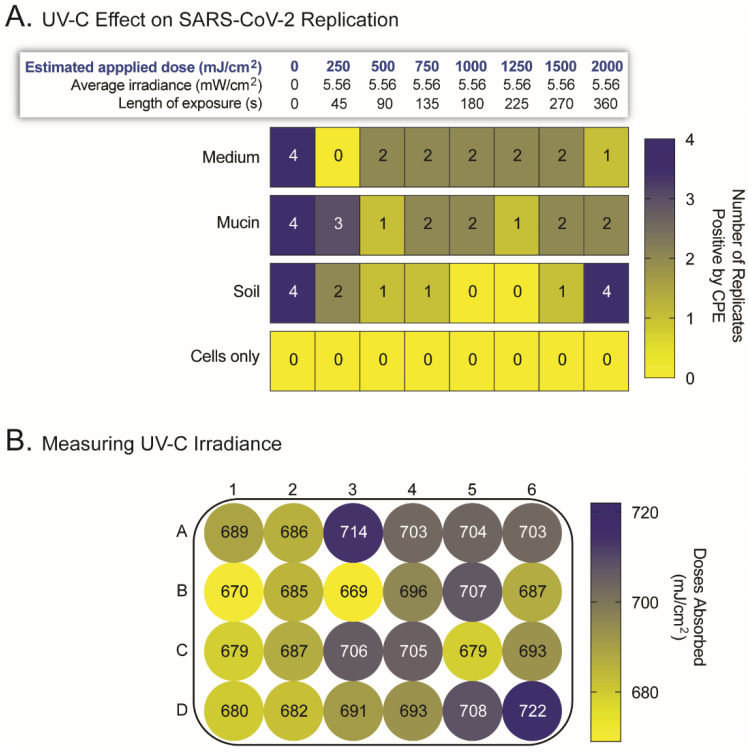
Use of UV-C irradiation to inactivate SARS-CoV-2 replication in stitched filtering facepiece respirator (FFR) coupons. (**A**) Stitched FFR coupons, four replicates per condition, were inoculated with 5 µL SARS-CoV-2 (10^7^ TCID_50_/mL) in a solution of cell-culture medium, mucin (simulating saliva), or soil load (mimicking respiratory secretions), then exposed to different applied doses of UV-C irradiation (250 to 2000 mJ/cm^2^). Following the treatment with UV-C, the FFR coupons were inverted so that the inoculated surface was in contact with the cell-culture medium in a 24-well plate containing 50,000 VeroE6/TMPRSS2 cells/well. Cells were incubated at 37 °C, 5% CO_2_ for 72 h and SARS-CoV-2 replication quantified by measuring cytopathic effect (CPE). Cells only were used as negative control, including untreated FFR coupons, with no SARS-CoV-2. (**B**) Quantification of the distribution of UV-C radiation across the UV-C chamber using chemical actinometry. The mean of the UV-C doses absorbed (mJ/cm^2^) from triplicate experiments for each position in a 24-well plate are indicated.

**Figure 2 pathogens-11-00083-f002:**
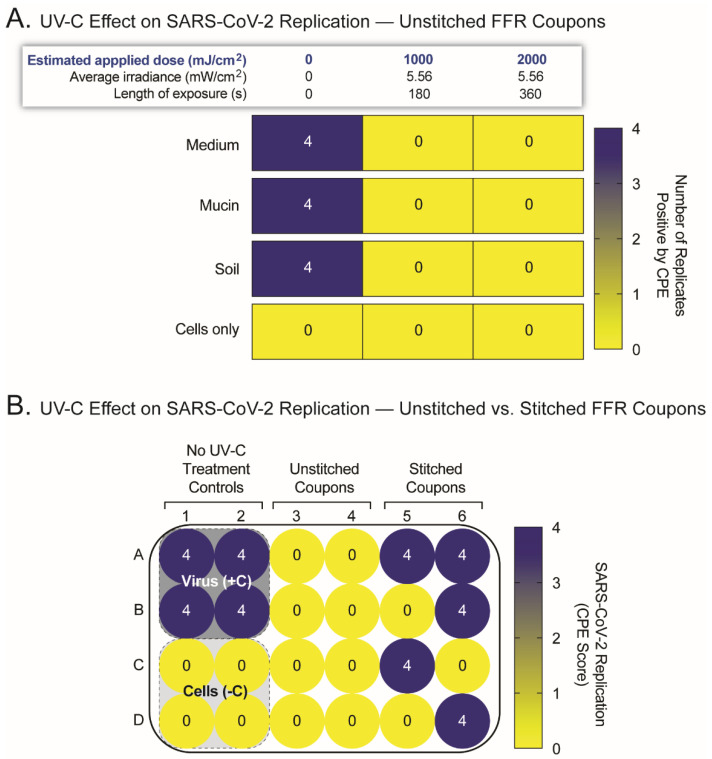
Use of UV-C irradiation to inactivate SARS-CoV-2 replication in unstitched filtering facepiece respirators (FFR) coupons. (**A**) Unstitched FFR coupons, four replicates per condition, were inoculated with 5 µL SARS-CoV-2 (10^7^ TCID_50_/mL) in a solution of cell-culture medium, mucin (simulating saliva), or soil load (mimicking respiratory secretions), then exposed or not to two applied doses of UV-C irradiation (1000 to 2000 mJ/cm^2^). FFR coupons were processed and analyzed as described in Figure 1A. (**B**) Head-to-head comparison of stitched vs. unstitched FFR coupons inoculated with SARS-CoV-2 and exposed to a single UV-C applied dose of 1000 mJ/cm^2^. Negative (cells only) and positive (cells plus SARS-CoV-2) controls correspond to FFR coupons not treated with UV-C irradiation. Cytopathic effect (CPE) score: 0, no CPE; 1, <25% CPE; 2, 25–49% CPE; 3, 50–74% CPE; and 4, 75–100% CPE.

**Figure 3 pathogens-11-00083-f003:**
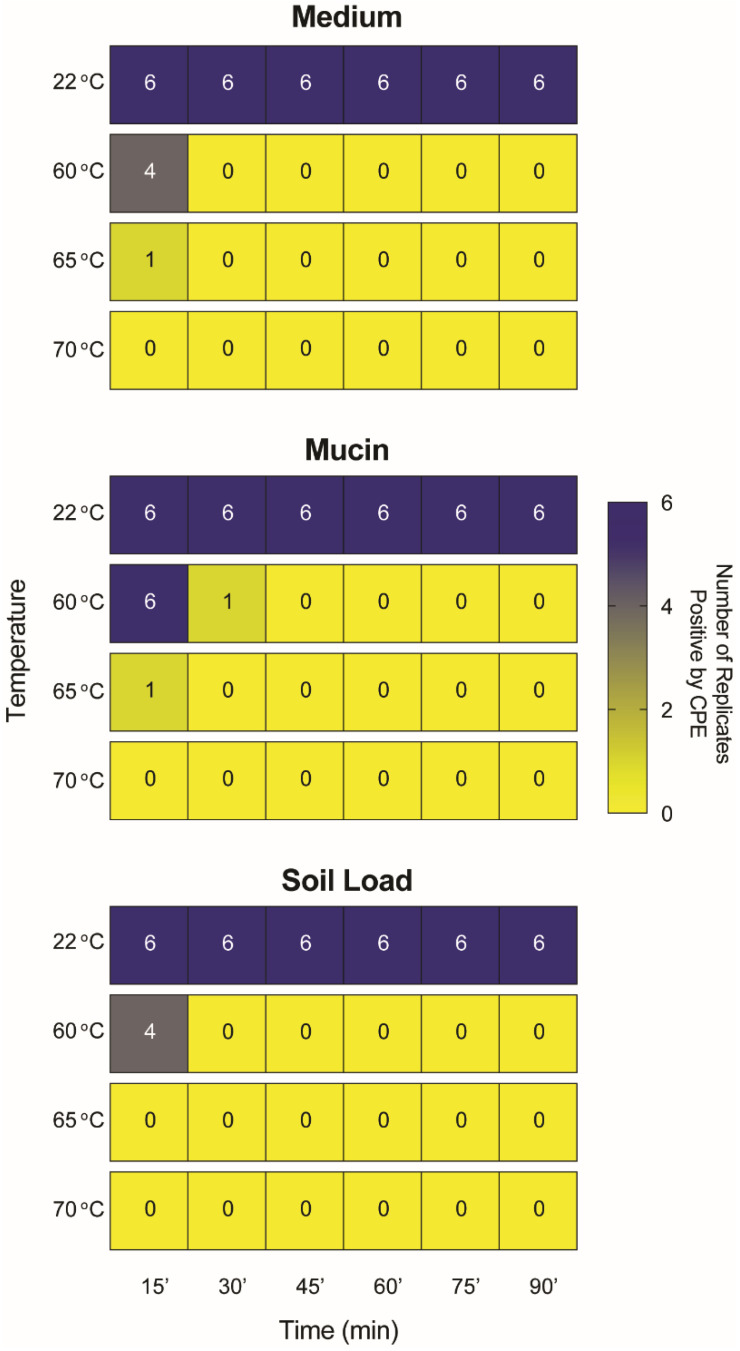
Use of dry heat to inactivate SARS-CoV-2 replication in stitched filtering facepiece respirators (FFR). Stitched FFR coupons, six replicates per condition, were inoculated with 5 µL SARS-CoV-2 (10^7^ TCID_50_/mL) in a solution of cell-culture medium, mucin (simulating saliva), or soil load (mimicking respiratory secretions), then incubated for 15 to 90 min at room temperature (approximately 22 °C), 60 °C, 65 °C, or 70 °C. Following the dry heat treatment, FFR coupons were processed and analyzed as described in Figure 1A. CPE, cytopathic effect.

**Figure 4 pathogens-11-00083-f004:**
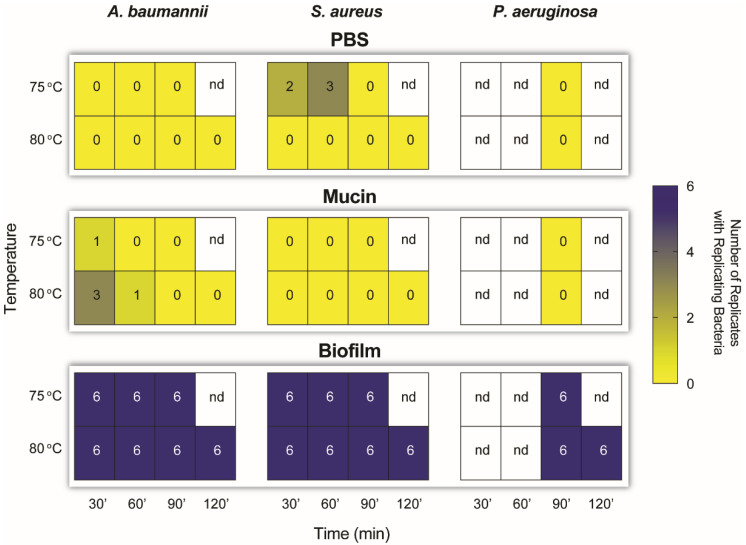
Inactivation of *Acinetobacter baumannii*, *Staphylococcus aureus*, and *Pseudomonas aeruginosa* on filtering facepiece respirator (FFR) coupons exposed to dry heat treatment. FFR coupons were inoculated with 1 × 10^6^ colony-forming units (CFU) bacteria in PBS, mucin (simulating saliva), or a colony biofilm with ~3.26 × 10^7^ to 4.4 × 10^8^ CFU. Dry heat treatment was performed at 75 °C and 80 °C for 30, 60, 90, or 120 min. Bacterial survival was determined by growth or no growth from the coupons and reported as the number of coupons (out of six replicates) with positive CFU. nd, not determined.

**Figure 5 pathogens-11-00083-f005:**
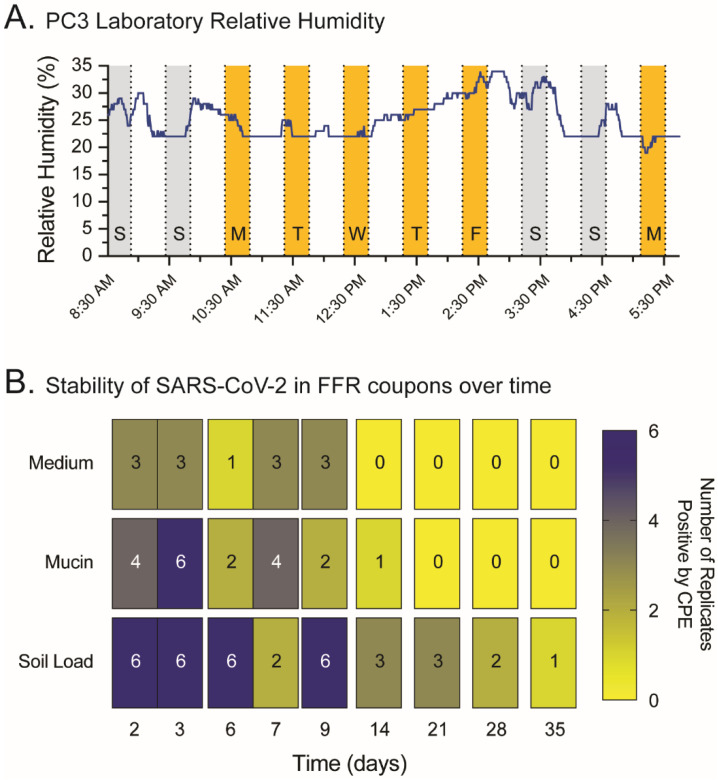
Stability of SARS-CoV-2 on filtering facepiece respirator (FFR) coupons over time. (**A**) The relative humidity in the PC3 laboratory was monitored during the first nine days of the viral stability experiment to establish the normal range of % humidity in the environment where the SARS-CoV-2-contaminated FFR coupons were stored. S, Sunday; M, Monday; T, Tuesday; W, Wednesday; T, Thursday; F, Friday; and S, Saturday. (**B**) Stitched FFR coupons, six replicates per condition, were inoculated with 5 µL SARS-CoV-2 (10^7^ TCID_50_/mL) in a solution of cell-culture medium, mucin (simulating saliva), or soil load (mimicking respiratory secretions), then stored for up to 35 days inside a Biosafety Cabinet in the PC3 laboratory, as described in Materials and Methods. FFR coupons were processed and analyzed as described in Figure 1A 2, 3, 6, 7, 9, 14, 21, 28, and 35 days post-inoculation with SARS-CoV-2. CPE, cytopathic effect.

## Data Availability

Raw data can be requested by contacting the corresponding authors.

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
