# Peer review of "Ultraviolet-C Irradiation, Heat, and Storage as Potential Methods of Inactivating SARS-CoV-2 and Bacterial Pathogens on Filtering Facepiece Respirators"

_pathogens, 2022, doi:10.3390/pathogens11010083_

Round 1

Reviewer 1 Report

Comments to Authors:

An important and elegant paper. Great that you tested whether you were able to rescue replication-competent virions, taking into account viral viability. Also positive that you reported challenges due to uneven surface texture and the presence of biological soil and biofilm on masks which effect results. Good that you that they tied the results of the stitching in the variability of the results after disinfection of the coupons could have implications for using UVGI on full masks.

While I think it’s interesting that SARS-CoV-2 in the soil load solution remained replication-competent in FFR for up to 35 days, or that better methods would be needed for heavily soiled PPE- it is crucial to mention that when speaking about reusing masks or PPE; visibly soiled masks would not be getting reused in the first place. Mask and PPE reprocessing is only for visually clean and undamaged articles- otherwise it is imperative to discard them.

Though you mention other types of decontamination like hydrogen peroxide (one could also mention ETO)- why is moist heat not mentioned or explored? Seeing as how other studies had more effective results with moist heat vs. dry heat, and it is a method that does not need much specialized equipment or chemicals, I think it could be useful to at least briefly explore this method perhaps citing a few key studies in the discussion section.

The study design/methods section is after the results, it would be nice to know how the study was conducted before we read the results; also some of the results section should be moved to the methods section. Similarly, the first part of the discussion section is better suited to the background section and the results sections.

Minor Comments

In Figures 3 and 4, I would recommend putting the time of exposure along the X axis for all three tables.

I am not sure that the graph in Figure 5A of relative humidity in the lab is especially useful for the paper, I think mentioning it in the text is sufficient.

Author Response

Response to Reviewer #1:

We appreciate the Reviewer’s comments, and thanking him/her for taking the time to read our “… important and elegant paper ...”. We appreciate his/her supportive comments and feedback about the design of our study. The reviewer also made some valid criticism, which we have addressed here and in the manuscript.

  1. While I think it’s interesting that SARS-CoV-2 in the soil load solution remained replication-competent in FFR for up to 35 days, or that better methods would be needed for heavily soiled PPE- it is crucial to mention that when speaking about reusing masks or PPE; visibly soiled masks would not be getting reused in the first place. Mask and PPE reprocessing is only for visually clean and undamaged articles- otherwise it is imperative to discard them.

We agree with the Reviewer and have included the following statement in the Discussion:

“These findings are important since in real-life, most bacteria grow in biofilms [69, 70] and we expect that this will be the case in heavily soiled PPE, which is not suitable for reuse for this reason. Mask and PPE reprocessing should only be for visually clean and undamaged articles - otherwise it is imperative to discard them. Other than discarding soiled PPE, further studies are necessary to explore the best treatment to completely eliminate viable bacteria biofilms using heat treatment from soiled material”

  1. Though you mention other types of decontamination like hydrogen peroxide (one could also mention ETO)- why is moist heat not mentioned or explored? Seeing as how other studies had more effective results with moist heat vs. dry heat, and it is a method that does not need much specialized equipment or chemicals, I think it could be useful to at least briefly explore this method perhaps citing a few key studies in the discussion section.

We agree with the Reviewer and have included the following statement in the Discussion:

“There is some evidence that non-steam moist heat treatment may be more effective for virus inactivation in FFRs than dry heat. However, dry heat treatment involves a much simpler protocol that would be markedly easier to replicate independently of scale or context, particularly at locations where funds are limited and/or access to more specialized equipment is difficult”

In addition, we believe that moist heat is harder to implement and control, and condensation of water within the system and on the PPE would be an added problem.  The PPE would possibly require an additional drying step with drier heat anyway.

  1. The study design/methods section is after the results, it would be nice to know how the study was conducted before we read the results; also some of the results section should be moved to the methods section. Similarly, the first part of the discussion section is better suited to the background section and the results sections.

Although we agree with the Reviewer, we are following the Guidelines for the Author from the journal. We will consult with the publisher to be sure that we can exchange the sections, including Materials & Methods before Results.

  1. In Figures 3 and 4, I would recommend putting the time of exposure along the X axis for all three tables.

We respectfully disagree with the Reviewer. Doing that will compromise the figure, making it too crowded and confusing with the titles of the respective tables. It is standard to use only one X axis when all the plots share the same values/units

  1. I am not sure that the graph in Figure 5A of relative humidity in the lab is especially useful for the paper, I think mentioning it in the text is sufficient.

We debated this internally and decided to keep Figure 5A in the paper. We believe it is important to show that relative humidity in the PC3 facility did not play a role in our findings.

Reviewer 2 Report

Dear Authors, 

Minor Comments:

  • Line 338-340: How relevant are sporforming pathogens (fungi, yeasts, etc.) regarding the reuse of face masks. How relevant are such pathogens for people with cystic fibrosis, immunodeficiencies, etc.? I suggest to mention the relevance of such pathogens, regardless if experiments with these pathogens have been performed within this study. It might be relevant for the reader to know why you have excluded spore forming pathogens in this study. 
  • Unfortunately I was unable to find whether the sewen FFR coupons have been treated with a disinfection measure before each experiment to guarantee their sterility? 

Author Response

Response to Reviewer #2:

We appreciate the comments, which we have addressed here and in the main text.

  1. Line 338-340: How relevant are sporforming pathogens (fungi, yeasts, etc.) regarding the reuse of face masks. How relevant are such pathogens for people with cystic fibrosis, immunodeficiencies, etc.? I suggest to mention the relevance of such pathogens, regardless if experiments with these pathogens have been performed within this study. It might be relevant for the reader to know why you have excluded spore forming pathogens in this study. 

We agree with the Reviewer; thus, we have included the following statement in the Discussion:

“Finally, we acknowledge the importance of spore-forming pathogens in the proposed scenario of PPE disinfection for potential reuse, and this is currently being studied (yeasts were not able to be studied in New Zealand but also are important to consider). From a practical perspective, clear policies regarding immediate discarding of PPE used in patients with vomiting or symptoms suggestive of pathogens such as Clostridium difficile, and not using reused PPE when caring for immunocompromised patients would be imperative”

  1. Unfortunately I was unable to find whether the sewen FFR coupons have been treated with a disinfection measure before each experiment to guarantee their sterility? 

FFR coupons were prepared as described in section 4.4 and Supp. Fig. 1. Neither the FFR material nor the synthetic threads were treated (disinfected) before the experiments since we thought that this could affect the integrity of the original material and findings. In the case of the experiments with viruses (IAV and SARS-CoV-2), the cell culture media contained standard concentration of antibiotics (penicillin and streptomycin), limiting the potential growth of bacteria.